# Rasch Validation of the VF-14 Scale of Vision-Specific Functioning in Greek Patients

**DOI:** 10.3390/ijerph18084254

**Published:** 2021-04-16

**Authors:** Ioanna Mylona, Vassilis Aletras, Nikolaos Ziakas, Ioannis Tsinopoulos

**Affiliations:** 12nd Department of Ophthalmology, Aristotle University of Medicine, Papageorgiou General Hospital of Thessaloniki, Agiou Pavlou 76, Pavlos Melas, 564 29 Thessaloniki, Greece; nikolasziakas@gmail.com (N.Z.); itsinop@med.auth.gr (I.T.); 2Department of Business Administration, University of Macedonia, 156 Egnatia str., 564 36 Thessaloniki, Greece; valetras@uom.edu.gr

**Keywords:** visual functioning, VF-14, Rasch modeling, PROM, HRQOL

## Abstract

The Visual Functioning-14 (VF-14) scale is the most widely employed index of vision-related functional impairment and serves as a patient-reported outcome measure in vision-specific quality of life. The purpose of this study is to rigorously examine and validate the VF-14 scale on a Greek population of ophthalmic patients employing Rasch measurement techniques. Two cohorts of patients were sampled in two waves. The first cohort included 150 cataract patients and the second 150 patients with other ophthalmic diseases. The patients were sampled first while pending surgical or other corrective therapy and two months after receiving therapy. The original 14-item VF-14 demonstrated poor measurement precision and disordered response category thresholds. A revised eight-item version, the VF-8G (‘G’ for ‘Greek’), was tested and confirmed for validity in the cataract research population. No differential functioning was reported for gender, age, and underlying disorder. Improvement in the revised scale correlated with improvement in the mental and physical component of the general health scale SF-36. In conclusion, our findings support the use of the revised form of the VF-14 for assessment of vision-specific functioning and quality of life improvement in populations with cataracts and other visual diseases than cataracts, a result that has not been statistically confirmed previously.

## 1. Introduction

The VF-14 scale was constructed by Steinberg et al. [1] as an index of functional impairment designed to serve a patient-reported outcome measure, originally for cataract patients. Patient-reported outcomes measures (PROMs) are the means of assessment that collect any information on patient-reported health, without external interpretation by a clinician or researcher and have gained a significant place in the assessment of treatment for ophthalmic diseases during the last two decades [2]. The reason for their proliferation is that they offer an unbiased indication of the real-life impact of the treatment on the patient’s life, a measure that can be weighed against the cost and burden of the treatment. Since a PROM is a self-report measure, rigorous validation is necessary to comprehensively and reliably assess the subjective experience of vision loss. The VF-14 scale has been employed extensively since its conception and its usage expanded in patient populations with other ophthalmic diseases. Its widespread adoption made it a perfect candidate for validity testing with sophisticated psychometric assessment methods such as Rasch analysis [3,4,5,6,7].

Rasch analysis is a probabilistic mathematical method that has been employed to assess the psychometric properties of a PROM instrument and its’ measurement quality against an established framework of precision criteria [8]. It transforms ordinal test responses into interval-level scores thereby reducing measurement noise, increasing precision, and statistical power to test the hypotheses with smaller sample size. Rasch analysis, therefore, has become a method of choice for examining the validity of a PROM instrument [9].

Previous examinations of the VF-14 with the employment of Rasch testing results were mixed, depending on the specific cultural characteristics of each population. The Chinese translated VF-14, VF-11R, and VF-8R were deemed valid and applicable [6]. In a German sample, the VF-14 was deemed adequate with minimal changes (collapse of two response categories for items 13 and 14) [10]. An attempt to validate a short nine-item version to an Asian population concluded that it did not have the range of items to assess the impact of vision impairment across the severity spectrum of vision loss [11].

The principal aim of this study is to assess a Greek version of the VF-14 using Rasch analysis and test its applicability in patients suffering from ophthalmic disease, including cataracts and other causes.

## 2. Materials and Methods

This was an observational prospective study of 300 patients who were treated for vision problems in the outpatient services of the 2nd Department of Ophthalmology, Aristotle University of Thessaloniki. The patients were longitudinally followed for two months, during which they received appropriate treatment depending on their underlying disease. The size of the research sample was determined by the statistical methods used. The results from the literature review on improvement in VF-14 scores post-surgery showed an improvement of at least one-third standard deviation from average before and after the intervention, meaning that a moderate effect size of 0.5 is reasonable [1,5,6,10]. To err on the side of caution, we assumed a conservative effect size (d) equal to 0.3 with a significance level (alpha) equal to 0.05 and a power index (beta) equal to 0.8. The required sample size equals 138 patients (or more). Power analysis for Rasch modeling relates to the modeled standard error (SE) of an item; if we seek a sample with 99% confidence that no item calibration is more than half a logit from its stable value, then the minimum sample size range is 108–243 test subjects depending on targeting, with recommended size at 150 test subjects [12]. Thus, we opted for a 300-patient sample with a sub-cohort of 150 patients with cataracts. This sub-cohort of 150 consecutive patients underwent phacoemulsification surgery and it was comprised of 86 men (57.3%) with a mean age of 73.84 years (SD = 8.55 years) and 64 women (42.7%) with a mean age of 73.45 years (SD = 7.05 years). A full list of the underlying disorders for the remainder of the sample is presented in Table 1.

The combined disorders group was comprised of 89 men (59.3%) with a mean age of 72.16 years (SD = 7.74 years) and 61 women (40.7%) with a mean age of 72.1 years (SD = 7.95 years). Exclusion criteria for all patients were the existence of other comorbid eye diseases, any complications related to their treatment, and any previous ophthalmic disease that is associated with low vision.

All patients were initially handed out a brief demographics questionnaire that included information on their gender, age, marital status, living arrangements, comorbid health issues that necessitated continuous medical care. The patients were required to fill in the Visual Function Index (VF-14) [1], a brief questionnaire designed to measure functional impairment on patients due to cataract, that has since been employed in various other ophthalmic diseases. It consists of 18 items (denoted as VF1-VF18 consistently in the manuscript) covering 14 aspects of visual function affected by eye disease. The difficulty undertaking each activity is rated on a five-category Likert scale ranging from zero for ‘Not possible’ to four for ‘no difficulty at all’ except for two items, items 13 and 14 which are rated on a four-category Likert scale ranging from one for ‘a lot of difficulty’ to four for ‘no difficulty at all’. This was the baseline measurement and a second measurement was carried out with the VF-14, two months after their first appointment. Additionally, their best-corrected visual acuity for the affected eye was also measured pre-and post-surgery with the Early Treatment Diabetic Retinopathy Study (ETDRS) charts. The patients with cataracts were additionally handed the Greek version of the Medical Outcomes Study Short-form 36 (SF-36) [13]. The SF-36 measures eight domains that are collapsed to create two distinct components: a physical dimension, represented by the Physical Component Summary (PCS), and a mental dimension, represented by the Mental Component Summary (MCS). PCS is composed of four scales assessing physical function, role limitations caused by physical problems, bodily pain, and general health. Higher scores represent better physical health. MCS is composed of four scales assessing vitality, social functioning, role limitations caused by emotional problems, and general mental health. Higher scores represent better mental health [14].

Although the VF-14 has been employed in studies with Greek patients before, it has not been statistically validated. There were six steps in the translation and cultural validation of the original English version following established guidelines [15]; concept elaboration, forward translation, back translation, proofreading, linguistic validation, content validation, and final validation.

A concept elaboration document was produced from all authors that included and compared results from other studies in Greek-speaking populations utilizing the same research instrument [16,17]. The translation process runs in parallel; two forward translations from English to Greek were independently conducted by the first two authors (I.M and I.T) who are medical doctors fluent in both languages with considerable experience with academic writing in English and experience working abroad. The other two authors (V.A and N.Z), who have considerable experience from scale-building and studied and worked in academia in the UK, each independently produced a backward translation. Each pair of authors gave feedback on the translations of the other pair and a single copy of forward and of a backward translation was produced. These copies were forwarded to a local translation service, external to the project, for a review of grammar and use of English and Greek. The results from the proofreading were returned for comparison and cross-check to produce a single acceptable version that was unanimously accepted and prepared for linguistic validation.

The linguistic validation was run under the supervision of the principal author who tested the draft in twenty patients for its comprehension and appropriateness. While the draft was well-received in terms of reading comprehension, it was determined that item appropriateness was very low for item 10 (“Taking part in sports, such as bowling, handball, tennis, golf”) since these sports are uncommon in the elderly Greek population. Bowling and golf have only a couple of venues in Greece in general while handball is practiced in a few organized sports clubs throughout Greece with limited participation to adolescents and young adults only. The twenty patients confirmed that they had never been involved with bowling, handball, or golf and that there was no opportunity to get involved with these activities had they wanted to in the past or now. Thus, in content validation that was approved by the unanimous decision of the authors, item 10 was changed to “Taking part in sports or exercising, such as running, fast strolling, playing soccer or tennis.” While running and fast strolling are not as dependent on the eyesight as bowling, handball, or golf, they still require a degree of visual attention to avoid injury. The change in content was cross-checked with the twenty patients who confirmed its appropriateness following a repeat proofreading from a bilingual individual (Greek-English) of mixed cultural heritage who was external to the project. The final validation of the draft was confirmed by all authors by unanimous decision.

### 2.1. Statistical Analysis

Gender differences in age and the VF-14 score were assessed with Mann-Whitney tests. The difference in VF-14 scores pre and post-operation was assessed with a paired samples t-test. All comparative statistics were calculated using the SPSS statistical package, version 25 (IBM Corp, Armonk, NY, USA). All subsequent Rasch measurements were carried out with the aid of the Winsteps^®^ Rasch measurement computer program [18]. Five fields of measurement were used to assess the validity of the Greek version of the VF-14 with Rasch modeling [4,19] including:

#### 2.1.1. Measurement Precision

Measurement precision refers to how the scale performs as an instrument of measurement. It is estimated with the person and item separation statistics. Separation is the signal-to-noise ratio in the data. Person separation indicates how efficiently a set of items can separate those persons measured, while item separation indicates how well a sample of people can separate those items used in the scale. A low person separation index (“PSI”) implies that the instrument may not be sensitive enough to distinguish between high and low performers, and more items may be needed while a low item separation index (“ISI”) implies that the person sample is not large enough to confirm the construct validity of the instrument [4]. A PSI of 1.5 represents an acceptable level of separation, an index of 2.00 represents a good level of separation, and an index of 3.00 represents an excellent level of separation [20]. A person separation index (PSI) of >2.0 and person reliability (PR) score of >0.8 are generally considered to be the minimum requirements for satisfactory discrimination of at least three strata of participants levels of the trait being investigated (i.e., vision functioning) [4,19].

#### 2.1.2. Unidimensionality

Unidimensionality is a prerequisite for construct validity since it refers to whether a scale measures only a single underlying trait (i.e., visual functioning), and it is assessed in Rasch measurement by examining the item fit statistics and with a principal component analysis (PCA) of the residuals. Item fit relates to how well the responses meet the test requirements and ultimately how well the items fit the construct. The item fit statistics are expressed in mean square statistics and there are two types of fit statistics, infit and outfit [4]. According to established criteria [7], mean fit values ranging between 0.5 and 1.5 are productive for measurement, values over 1.5 are unproductive for construction of measurement, but not degrading, values under 0.5 are less productive for measurement, but not degrading and values over 2 denote an item that distorts or degrades the measurement system. To test for local independence the method of choice is the conduct of a PCA of the residuals, a process in which we scan for patterns in the part of the data that does not accord with the Rasch measures. If this is the case, then there is a possibility that a second dimension is present that may distort measurement and the unidimensionality criterion is not upheld. When 60% of the variance in the PCA of the residuals is explained by the raw data then this is an indication of unidimensionality since there is little noise to form a pattern [19]. Residuals in PCA are grouped in contrasts and if the first contrast has an eigenvalue of >2.0, then this is considered as evidence that a second contrast is being measured by the scale [19].

#### 2.1.3. Category Threshold Order

The response categories for the items in a scale should ideally be used in an orderly fashion. This requires that the category definitions are clear and distinct to one another and the number does not exceed the range that the respondents can distinguish or is smaller than the nuances of the category that we are trying to ascertain [21]. If there is disordering, then some answers are significantly more likely than others or even unlikely.

#### 2.1.4. Targeting

Targeting refers to how far the average or modal measure is from the center of the item calibrations, denoting how persons of higher or lower ability (i.e., visual functioning) will be able to relate to the items that are offered and respond meaningfully [3]. Perfect targeting would have a difference in means equal to zero logits and poor targeting over two logits, while a value between 0.5 and 1 logit indicates very good targeting [22].

#### 2.1.5. Differential Item Functioning

Differential item functioning (DIF) indicates whether subgroups are responding in a different pattern than the rest of the sample despite having equal levels of the assessed trait [4]. To ascertain clinically important differential item functioning, two conditions had to be satisfied at the same time: a Welsh’s test statistically significant *p*-value (*p* < 0.05) and a contrast value of >0.64 logits. If both conditions were satisfied it would indicate that the interpretation of the scale differs by group and that it is influenced by confounding factor(s).

## 3. Results

### 3.1. Rasch Analysis

#### 3.1.1. Measurement Precision

In our sample, the VF-14 scale had a PSI = 2.06 and a PR = 0.81, which were satisfactory values. However, the VF-14 showed a poor result in the fit statistics with a large number of items exhibiting MSNQ higher than 1.5. The PCA had 61.2% of raw variance explained by the measures but the unexplained variance by the first contrast of the residuals was 2.46 eigenvalue units for the full scale and there was a second contrast with 2.35 units. As a result, an alternate version was created with 8 items, which will be referred to as the revised Greek version of the Visual Function scale, ‘VF-8G’. All MSNQ values of the revised version adhered to the guidelines that were mentioned (Table 2). The PCA of the revised VF-8G had 64.6% of raw variance explained by the measures while the unexplained variance by the first contrast of the residuals was 1.99 eigenvalue units, demonstrating better unidimensionality than the VF-14. The VF-8G has a PSI = 2.85 and a PR = 0.89, showing better metrics than the original VF-14.

#### 3.1.2. Category Threshold Order

The original version of the VF-14 had notably disordered category probabilities with the answer “yes, with a great deal of difficulty” being completely improbable in any item measure. In contrast, the revised VF-8G had a more smoothly transitioning category probabilities map, with an increased probability for the first and last response categories, depending on the person item measure (Figure 1).

#### 3.1.3. Targeting

Both versions of the VF-14 had acceptable targeting, the revised VF-14 had a difference between the person and item means on the person-item map equal to −0.44 while the revised VF-14 had 0.68.

#### 3.1.4. Differential Item Functioning

Differential item functioning for gender, age, and underlying disorder was examined for the VF-8G. Gender was included because there are differences between the genders with regards to the usual activities that they perform and value the most; hence, potentially, they would place a differential emphasis on the items of the scale that were more closely related to their everyday needs. Age has a direct impact on visual functioning but also the activities that the patients are expected to perform since the higher the age the more likely the chance of comorbid disease that limits general functionality. We divided the sample into two subsamples for this DIF analysis, those patients up to and including 70 years of age, since they comprised one-third of the total sample and those aged over 70. DIF for the underlying disease was examined since the VF-14 scale originally was for cataract patients; hence, we divided the sample into two subsamples, cataract patients and those patients with any other underlying disease

Table 3 presents the summary of the examination of the VF-8R items for differential item functioning by gender, age, and disorder (cataract or other).

Results indicated that there was a single item in each instance that met the statistical significance for differential functioning (Welch’s test *p* < 0.05), but in every case, the contrast effect size was lower than 0.64 denoting that the difference in functioning between the subgroups was not meaningful. These items were item 1 for gender, item 2 for age, and item 12 for the underlying disorder.

#### 3.1.5. Person-Item Map

There are two person-item maps presented, Figure 2a belonging to the original VF-14 and Figure 2b to the revised VF-8G scale.

Each person-item map displays the participant scores on the Rasch-calibrated scale and the relative difficulty of each of the scale items. On the left side of each Wright Map, there are the mean (M) and two standard deviation points (S = one SD and T = two SD) for each patient’s vision functioning. Participants with the highest level of vision functioning are located at the top of the figure while those with the lowest vision function are found at the bottom. On the right side of the map, the mean difficulty of the items (M) and two standard deviation points (S = one SD and T = two SD) for the items are shown, where ‘mean difficulty’ refers to the mean possibility of answering positively the item, an item being ‘more difficult’ when fewer participants answer it positively. In the case of the original VF-14 scale (Figure 2a), several items (VF10, VF17, VF15, and VF14) relate to very few patients, whereas there is a significant overlap in ability between items VF2 and VF7, VF6, and VF8 denoting redundancy. In the Wright map of the revised VF-8G (Figure 2b), there is a better spacing between the items denoting little redundancy, and more discriminate ability between the items with the person ability ranging from −6 to 7 logits compared to the range −3 to +3. A relative weakness of the revised VF-8G is that there is a lack of items to target participants at the higher end of the scale (i.e., those with more visual functioning) since most items were too easy to perform for those patients. This leads to the finding that the mean (M) ability of the patients is higher than the mean (M) difficulty of the items. However, since this difference is less than one standard deviation, this is not a significant issue.

Table 4 presents a comparative summary between the original VF-14 scales in English and Greek and the proposed eight-item versions for both languages, VF-8R and VF-8G respectively.

Table 5 presents the results in logits from the application of the VF-8G scale into the sample, per disease, and gender.

Results indicate that the cataract group had statistically significantly lower visual functioning than the combined group of other diseases, Mann-Whitney Z = 2.717, *p* = 0.007, while there was no difference in visual functioning between the genders in either sub-group (Mann-Whitney Z = 1.778, *p* = 0.075 for the cataract group and Z = 1.639, *p* = 0.101 for the combined disorders group). These comparisons are examples of how Rasch scoring can assist in a typical clinical setting since the ordering of all patients in the same axis regardless of the underlying disorder and gender thereby permitting us to exclude valuable comparative information.

### 3.2. Additional Examinations of the VF-8G Reliability, Content, and Concurrent Validity

The reliability of the VF-8G is assessed with two measurements, Cronbach alpha’s score for the VF-8G equals 0.9, while the more accurate Rasch measurement methodology offers a model reliability upper estimate of 0.91 and a ‘real’ reliability lower estimate of 0.89. In every case, the reliability of the VF-8G is excellent.

We examined the difference in VF-8G scores pre-and post-surgery in the cataract patients’ group, assuming that corrective surgery would carry a positive effect on the visual functioning of the patient to test content validity. A paired-sample t-test returned a statistically significant difference between visual functioning pre and post cataract surgery assessed with the VF-8G, t (149) = 17.684, *p* < 0.001. To ascertain concurrent validity, we examined the correlation between the scores on the VF-8G and the visual acuity pre-and post-surgery was examined, and results indicated that the VF-8G score after surgery correlated with the improvement between visual acuity pre-and post-surgery, Spearman’s rho r(s) = 0.161, *p* < 0.05.

Additionally, we compared this new Greek version against the proposed eight-item version (VF-8R) put forward by Gothwal et al. [5]. The correlation between the scores from the VF-8G and the VF-8R scales was tested with the Spearman correlation coefficient. The result was highly correlated both in the preoperative measurement (r_s_ = 0.936, *p* < 0.001) and the post-operative measurement (r_s_ = 0.954, *p* < 0.001) The mean difference between the two versions for the preoperative measurements was −1.19 points (C.I −1.409 to −0.97) and for the postoperative measurements −1.43 points (C.I −1.641 to −1.224) *p* < 0.001 in both cases. The two Bland-Altman 95% plots describe graphically the measure of agreement between the two versions with 95% confidence intervals (Figure 3a,b). Few outliers were noted outside the confidence intervals.

Lastly, as an additional measure of convergent validity, the difference in VF-8G scores before and after the surgery in cataract patients was correlated to the corresponding difference in the two components of the SF-36 scale; Spearman r_s_ = 0.432, *p* < 0.001 for the MCS and Spearman r_s_ = 0.196, *p* = 0.016 for the PCS. These correlations were comparable and slightly more favorable to those of the full version with the 14 items (Spearman r_s_ = 0.372, *p* < 0.001 for the MCS and Spearman r_s_ = 0.164, *p* = 0.045 for the PCS).

An Excel file that can be used to transform test scores to Rasch logits directly is offered as Appendix A, the user entering the numerical values 0 to 4 in the ‘Patient scores’ sheet and reading the transformed scores in the ‘Converted scores’ sheet.

## 4. Discussion

As with previous validation studies of the VF-14 with the employment of Rasch testing [6], cultural effects were significant in our Greek sample as well, leading to the formation of a smaller scale with 8 items, since the original scale demonstrated poor unidimensionality and low targeting of items. The revised eight-item version had solid metrics and it has the benefit of simplicity over the full 14-item version. This version has different items compared to the VF-8R previously validated in American and Chinese populations, denoting significant cultural differences between the populations in question. The Greek revised VF-14G includes items 1–4, 6–8, and 12 from the original version. The removed items featured questions on difficulty noticing steps, playing card and board games, engaging in physical outdoor activities, cooking, and driving. These omissions reflect a difference in everyday routines between Greek and other populations of elderly patients; typically, Greek elders live in the context of an extended family where they are assisted with outdoor chores and obligations and keeping mostly indoors; thus, outdoor leisure activities and driving a car is less common. According to the latest data of the Hellenic Statistical Authority [23], one out of four families is an extended one, directly including the elder grandparents, while only a small percentage of elders live in elderly homes, instead of living under the close supervision of their adult children. Cooking is typically considered a housewife’s obligation hence it was expected that item 11 would not perform well in a mixed-gender sample. The items included in the VF-8G mostly refer to a quiet and reserved lifestyle with a limited need for self-reliance.

A limitation of this study is the inability to provide quality-adjusted life-year utility values, a process that has been practiced elsewhere [24]. The Greek SF-36 has not been Rasch-tested, an official Greek SF-6D version does not exist and it is unclear whether the modification would be appropriate for the Greek-speaking population. Validity testing for the Greek SF-36 itself has shown that a three-factor second-order model was more plausible than the two-factor second-order one [25]. The process that is employed to generate an SF-6D utility score demands the computation of preference weights but unfortunately, there is no valuation survey completed or currently underway in Greece. The only set of preference weights currently available are from a UK representative sample [26] and since two countries who are culturally more similar to Greece than the UK (Portugal, Spain) have opted to produce their own sets of weights, the UK sets will likely be inappropriate for use with the Greek population. Hence, we did not employ the SF-36 scores further in this validation study.

This study is the first to provide a validated version of the VF-14 for use in Greek-speaking populations. Its strength lies in the appropriateness of the Rasch method for examining scale reliability and validity. While several previous studies employed the VF-14 in Greek-speaking populations [16,17], their validity is essentially unknown. In the content validation process, it was immediately clear that item 10 of the original scale should have been amended to be culturally appropriate while even the validity of the amended VF-14, in general, was problematic. The eight-item version is shorter and easier to deploy, meanwhile, any author can employ the Rasch weights that we have made available in the Appendix A. The study sample also included patients with various other eye diseases and statistically confirmed the appropriateness of the new version. This is contrary to other validation studies that have not examined whether the VF-14 is appropriate for use in patient samples other than those suffering from cataracts. Since the VF-14 is used regardless of this omission in other patient groups as well, we consider this as an important step in scale validation in this particular case. A limitation of our study is that all patients originated from a single center. However, since this is a tertiary center of care with a wide epidemiological catchment area including both metropolitan and rural areas, we consider our sample as indicative of the Greek patient population at large.

## 5. Conclusions

In conclusion, our validation study has resulted in a rigorously tested shortened version of the VF-14, the VF-8G, better suited for Greek-speaking populations. Findings also support the use of the VF-8G in populations with other visual diseases than cataract, the original patient group for the VF-14 scale, a finding that has not been statistically confirmed previously.

## Figures and Tables

**Figure 1 ijerph-18-04254-f001:**
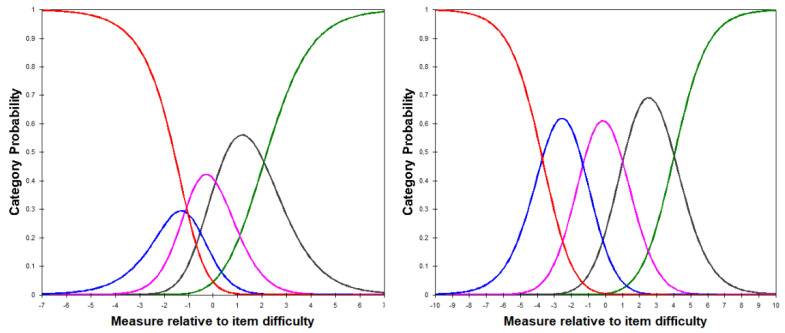
Category probability curves for the VF-14 (left) and the VF-8G scales, demonstrating the operation of the five-item Likert-style response categories. The VF-14 has clearly disordered thresholds with the second response lacking any range along with the ability score where it is most likely to be chosen over the other responses.

**Figure 2 ijerph-18-04254-f002:**
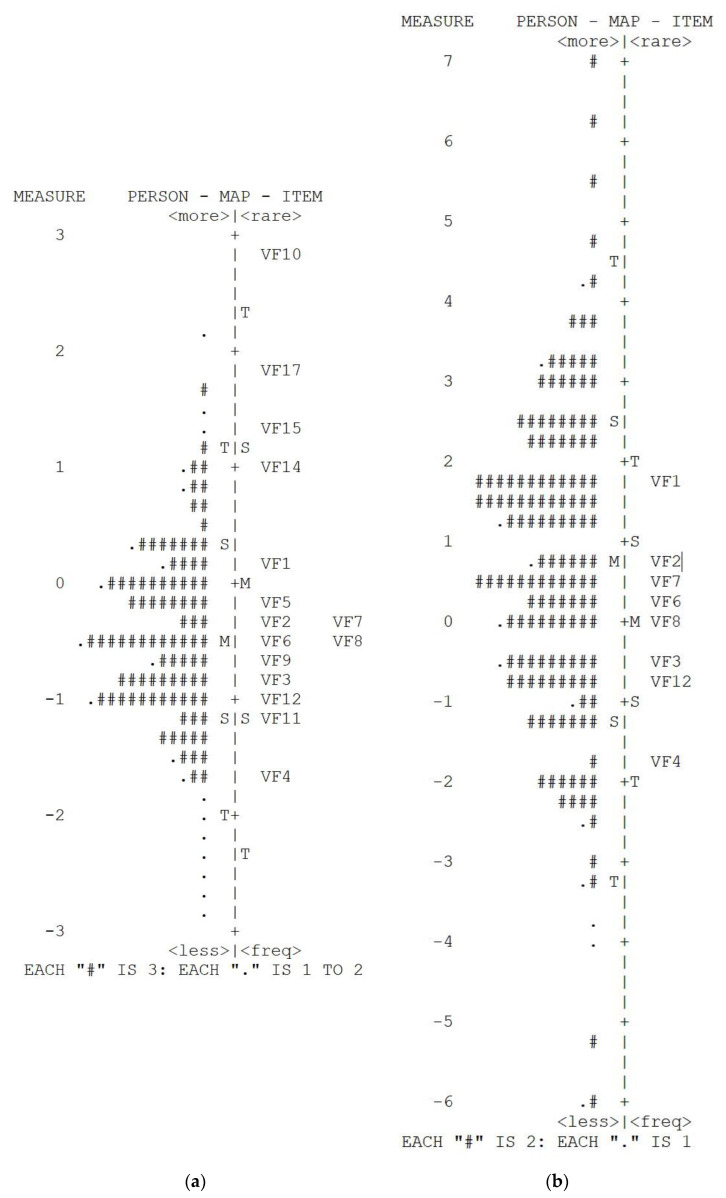
(**a**). Wright map for the VF-14 scale. Items on the scale are marked VF1-VF17, M = mean difficulty of the items, S = one standard deviation, T = two standard deviations; (**b**). Wright map for the VF-8G scale. Items on the scale are marked VF1-VF12, M = mean difficulty of the items, S = one standard deviation, T = two standard deviations.

**Figure 3 ijerph-18-04254-f003:**
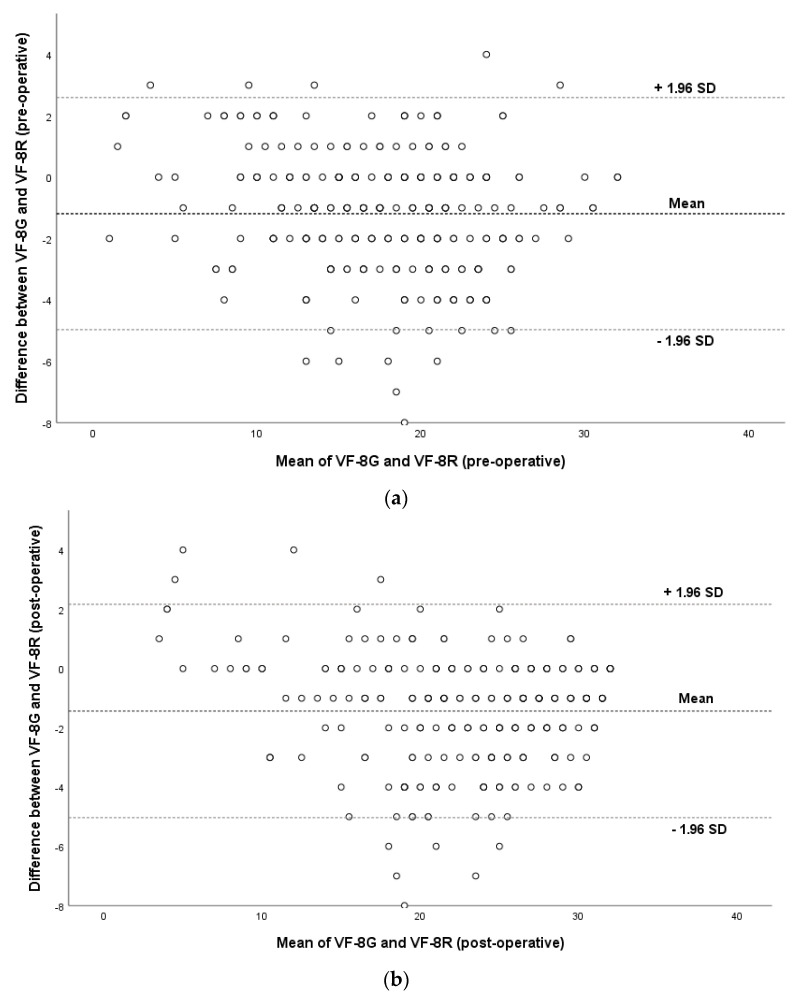
(**a**). Bland-Altman plot of agreement between the Gothwal et al. [5] 8-item VF-8R and the Greek version VF-8G pre-operatively. (**b**). Bland-Altman plot of agreement between the Gothwal et al. [5] 8-item VF-8R and the Greek version VF-8G post-operatively.

**Table 1 ijerph-18-04254-t001:** Underlying disorders.

Disorder	N	Percent	Age in Years (Mean/SD)	Males (N/%)	Females (N/%)
Cataract	150	50	73.67 (7.93)	86 (57.3%)	64 (42.7%)
Age-related macular degeneration	30	10	74.13 (6.23)	18 (60%)	12 (40%)
Glaucoma	24	8	69.04 (8.47)	13 (54.2%)	11 (45.8%)
Ectropion	18	6	71 (7.5)	15 (83.3%)	3 (16.7%)
Proliferative diabetic retinopathy	16	5.3	67.56 (10.16)	11 (68.8%)	3 (31.3%)
Canalicular obstruction	13	4.3	73.46 (6.1)	5 (38.5%)	8 (61.5%)
Blepharitis	13	4.3	74.92 (5.34)	7 (53.8%)	6 (46.2%)
Central vein occlusion	10	3.3	72.8 (4.1)	3 (30%)	7 (70%)
Dry eye	10	3.3	72.6 (6.77)	8 (80%)	2 (20%)
Retinal vein occlusion	5	1.7	69 (4.06)	2 (40%)	3 (60%)
Ptosis	5	1.7	71.6 (4.56)	3 (60%)	2 (40%)
Keratoconus	4	1.3	61.5 (2.88)	4 (100%)	
Fuchs’s endothelial dystrophy	2	0.7	51 (2.82)		2 (100%)
Total	300	100			

**Table 2 ijerph-18-04254-t002:** Fit statistics for the modified VF-14 items.

Original VF-14 Item	MODEL		INFIT		OUTFIT		EXACT MATCH	
	Measure	S.E.	MNSQ	ZSTD	MNSQ	ZSTD	Observed%	Expected%
VF6	0.33	0.09	1.51	5.50	1.51	5.29	53.7	58.9
VF4	−1.86	0.10	1.26	3.00	1.28	2.79	53.7	62.3
VF1	1.72	0.09	1.03	0.37	1.02	0.20	52.7	57.9
VF12	−0.80	0.10	0.94	−0.74	0.95	−0.55	63.4	60.3
VF3	−0.58	0.10	0.87	−1.65	0.86	−1.75	69.8	59.7
VF8	0.11	0.10	0.82	−2.34	0.82	−2.22	63.4	59.2
VF7	0.43	0.09	0.80	−2.60	0.78	−2.87	64.1	58.4
VF2	0.66	0.09	0.74	−3.57	0.74	−3.38	61.4	58.1
Mean	0.00	0.10	1.00	−0.3	0.99	−0.3	60.3	59.3
P.SD	1.01	0.00	0.25	2.9	0.25	2.8	5.8	1.3

S.E. = Standard Error, MSNQ = Mean Square, ZSTD = Z–standardized, P.SD = Population Standard Deviation.

**Table 3 ijerph-18-04254-t003:** Differential Item Functioning (DIF) by Gender, age, and disorder.

Original VF-14 Item	DIF by Gender				DIF by Age				DIF by Disorder			
	Male	Female	Contrast	Welch’s Test *p*-Value	≤70 Years	>70 Years	Contrast	Welch’s Test *p*-Value	Cataract	Other	Contrast	Welch’s Test *p*-Value
VF1	1.89	1.47	0.42	0.027	1.82	1.67	0.15	0.462	1.85	1.59	0.26	0.174
VF2	0.66	0.66	0.00	1.000	0.97	0.50	0.46	0.02	0.66	0.66	0.00	1.000
VF3	−0.52	−0.68	0.17	0.397	−0.54	−0.61	0.07	0.736	−0.58	−0.58	0.00	1.000
VF4	−1.91	−1.79	−0.12	0.565	−2.09	−1.75	−0.33	0.131	−1.86	−1.86	0.00	1.000
VF6	0.18	0.54	−0.36	0.06	0.42	0.29	0.13	0.513	0.48	0.17	0.31	0.102
VF7	0.47	0.37	0.10	0.6	0.39	0.43	−0.04	0.848	0.47	0.39	0.07	0.692
VF8	0.04	0.20	−0.16	0.41	−0.04	0.18	−0.22	0.284	−0.01	0.23	−0.24	0.207
VF12	−0.83	−0.77	−0.05	0.787	−1.00	−0.71	−0.29	0.162	−1.00	−0.60	−0.40	0.041

DIF = Differential Item Functioning.

**Table 4 ijerph-18-04254-t004:** Rasch-based metric properties of the English and Greek VF-14 scales and the short versions proposed by Gothwal et al. [5] and this study (VF-8G).

Parameter	English Version		Greek Version	
	VF-14	VF-8R	VF-14	VF-8G
Number of items	14	8	14	8
Measurement precision	2.45	2.29	2.06	2.85
Mis-fitting items	2	0	6	0
Mean person location	−1.86	−1.97	−0.44	0.68
PCA, eigenvalue first contrast	2.3	1.6	2.46	1.99

**Table 5 ijerph-18-04254-t005:** Results from the application of the VF-8G scale in the research sample.

Estimated Person Measure (Mean/S.D)		Gender		Total
		Male	Female	
Disease Group	Cataract	0.464 (2.12)	0.05 (1.77)	0.29 (1.98)
	Other	0.921 (1.95)	1.274 (2.27)	1.06 (2.08)
	All	0.696 (2.04)	0.65 (2.11)	

## Data Availability

Data available on request for scientific reasons due to restrictions placed by the Institutional Review Board that approved the study for reasons of privacy.

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
