# Peer review of "Rasch Validation of the VF-14 Scale of Vision-Specific Functioning in Greek Patients"

_ijerph, 2021, doi:10.3390/ijerph18084254_

Round 1
Reviewer 1 Report
The manuscript details the validation of the VF-14 scale of vision-specific functioning in Greek patients using Rasch modeling. The introduction is clear, the concepts behind the validation are carefully explained and the results are well presented.
For further improvement, I suggest the authors explain how they calculated the required sample size (150 patients with cataract and 150 with other diseases), so it can be useful to readers who may want to do a similar analysis in the future.
Methods, page 3, the abbreviations “PSI” and “ISI” should be between parenthesis and explained when used for the first time.
Although both groups were sampled in two waves (initially and after surgery or procedure), validity was only assessed with the VF-8 in the cataract group. It would be better to present the results for both groups or, if not performed in the “other etiologies” group, modify the abstract accordingly (instead of “A revised eight-item version, the VF-8G, was tested and confirmed for validity in the research population” writing “A revised eight-item version, the VF-8G, was tested and confirmed for validity in the cataract research population”).
Overall, the manuscript is very easy to read and I congratulate the authors for their care in adapting the questions to the Greek population, such as those reflecting the life in extended families and the gender-associated “obligations” in their elderly society.
Reviewer 2 Report
This study employed a Rasch analysis of the VF-14 scale in a Greek population. It recommended changes based on the results. The modified scale was assessed in two clinical populations.
The methods were very straight forward, very similar to similar studies in other populations (Chinese and German). The statistical analysis seemed straight forward and appropriate with one minor exception (see below). The results/conlusions of the modified scale had face validity from cultural expectations.
The manuscript was well written, organized, and clear.
Two minor issues were noted. Even thought the "sports" may have been more culturally appropriate, I found it a bit strange that the authors chose to replace some visually dependent sports (bowling, handball) from the VF-14 with much less visual dependent sports (running, and fast strolling). This was even more unusual because one of the cultural arguments used to explain the results was that the population that tended to go outdoors much.
Another analysis common in these Rasch studies with the VF-14 is a Bland-Altman analysis that was not included.
